# Nutritional Enrichment of Plant Leaves by Combining Genes Promoting Tocopherol Biosynthesis and Storage

**DOI:** 10.3390/metabo13020193

**Published:** 2023-01-28

**Authors:** Luca Morelli, Laura García Romañach, Gaetan Glauser, Venkatasalam Shanmugabalaji, Felix Kessler, Manuel Rodriguez-Concepcion

**Affiliations:** 1Institute for Plant Molecular and Cell Biology (IBMCP), CSIC-Universitat Politècnica de València, 46022 Valencia, Spain; 2Centre for Research in Agricultural Genomics (CRAG) CSIC-IRTA-UAB-UB, Campus UAB Bellaterra, 08193 Barcelona, Spain; 3Neuchâtel Platform of Analytical Chemistry, Faculty of Sciences, University of Neuchâtel, 2000 Neuchâtel, Switzerland; 4Laboratory of Plant Physiology, Faculty of Sciences, University of Neuchâtel, 2000 Neuchâtel, Switzerland

**Keywords:** tocopherols, vitamins, biofortification, *N. benthamiana*, transient expression, chromoplasts

## Abstract

The enrichment of plant tissues in tocochromanols (tocopherols and tocotrienols) is an important biotechnological goal due to their vitamin E and antioxidant properties. Improvements based on stimulating tocochromanol biosynthesis have repeatedly been achieved, however, enhancing sequestering and storage in plant plastids remains virtually unexplored. We previously showed that leaf chloroplasts can be converted into artificial chromoplasts with a proliferation of plastoglobules by overexpression of the bacterial *crtB* gene. Here we combined coexpression of *crtB* with genes involved in tocopherol biosynthesis to investigate the potential of artificial leaf chromoplasts for vitamin E accumulation in *Nicotiana benthamiana* leaves. We show that this combination improves tocopherol levels compared to controls without crtB and confirm that *VTE1*, *VTE5*, *VTE6* and *tyrA* genes are useful to increase the total tocopherol levels, while *VTE4* further leads to enrichment in α-tocopherol (the tocochromanol showing highest vitamin E activity). Additionally, we show that treatments that further promote plastoglobule formation (e.g., exposure to intense light or dark-induced senescence) result in even higher improvements in the tocopherol content of the leaves. An added advantage of our strategy is that it also results in increased levels of other related plastidial isoprenoids such as carotenoids (provitamin A) and phylloquinones (vitamin K1).

## 1. Introduction

The demand for plant products enriched in metabolites with vitamin activity has dramatically increased in the last few years. Plant-produced vitamins include several of isoprenoid origin such as carotenoids (particularly, provitamin A β-carotene), tocochromanols (vitamin E) and phylloquinone (vitamin K1). In particular, the term: “Vitamin E” (VitE) refers to four tocopherols and four tocotrienols classified as α-, β-, γ-, and δ- forms. These tocochromanols are only synthesized in photosynthetic organisms such as cyanobacteria, algae, and plants, while animals must assimilate them through their diet [1,2,3]. 

VitE has an important antioxidant role because it buffers reactive oxygen species (ROS) by quenching free radicals and protecting against lipid peroxidation [4,5]. Furthermore, VitE contributes to the inhibition of platelet aggregation, inhibition of cell proliferation, monocyte adhesion, immune system regulation, and protection against DNA damage, thus alleviating several pathologies such as infarction and neurodegenerative disorders (e.g., Alzheimer’s Disease) [4,6,7]. For this reason, tocochromanols are widely consumed around the world, especially in plant-derived food sources that mainly contain α-tocopherol (α-TC) and γ-tocopherol (γ-TC). The first one (specifically, the stereoisomer RRR-α-tocopherol) is the one with the highest biological VitE activity since it is selectively retained by the hepatic α-TC transfer protein, whereas the other VitE forms are usually metabolized by the liver, resulting in a generally lower biological activity [3,5].

In plants, α-TC is typically abundant in photosynthetic tissues, whereas γ-TC and tocotrienols are found in higher amounts in seeds, fruits, and nuts. All plant tocochromanols are produced in plastids such as chloroplasts of green tissues, chromoplasts of carotenoid-accumulating flowers and fruits, amyloplasts of seeds and tubers, and leucoplasts of white petals [3,8]. The amount of these metabolites, however, differs with the plastid type and with the species and tissue. Chromoplasts of tomatoes, red peppers, and daffodil petals, for example, are reported to contain moderate levels of α-TC and γ-TC while artificially obtained leaf chromoplasts were shown to accumulate up to twofold more tocopherols than green leaf chloroplasts [9,10]. 

Tocopherols and tocotrienols share many enzymes and substrates in their biosynthetic pathways (Figure 1). They contain a polar head group derived from homogentisic acid (HGA), and a hydrophobic tail synthesized from phytyl-diphosphate (phytyl-PP) in the case of tocopherols and geranylgeranyl diphosphate (GGPP) in the case of tocotrienols [1,3]. HGA is produced by the enzyme p-hydroxyphenyl pyruvate (HPP) through dioxygenase (HPPD, encoded by the *PDS1* gene in *Arabidopsis thaliana*) [11]. HPP is typically formed from chorismate via prephenate, arogenate, and tyrosine (Tyr) by the action of arogenate dehydrogenase that, in plants, is feedback inhibited by high Tyr levels [12]. On the other hand, the saturated tail of tocopherols is derived from phytyl-PP obtained: (a) from GGPP via GGPP reductase (GGR) or (b) from phytol released from chlorophyll by phytol kinase (PHK, encoded by the *VTE5* gene) and phosphorylated in the stroma by a phytyl-P kinase (PHPK, encoded by the *VTE6* gene) [3,11,13]. HGA and phytyl-PP are joined by homogentisate phytyltransferase (HPT) encoded by *VTE2* to form 2-methyl-6-phytyl-1,4-benzoquinol (MPBQ) which is converted into 2,3-dimethyl-6-phytyl-1,4-benzoquinone (dMPBQ) by MPBQ methyltransferase (MT, encoded by *VTE3*) or into δ-TC by tocopherol cyclase (TC, encoded by *VTE1,* responsible also for the synthesis of γ-TC from dMPBQ) [12,14]. Finally, γ-TC methyltransferase (γ-TMT, encoded by *VTE4*) converts δ-TC and γ-TC to β-TC and α-TC, respectively. In the case of tocotrienols, homogentisate geranylgeranyl transferase (HGGT) condenses GGPP and HGA to produce 2-methyl-6-geranylgeranyl-1,4-benzoquinol (MGGBQ) that is then converted to 2-dimethyl-6-geranylgeranyl-1,4-benzoquinol (dMGGBQ) by a methyltransferase. At this point, the VTE1 and VTE4 enzymes use geranylgeranylated intermediates as substrates to produce tocotrienols (Figure 1). The availability of phytyl-PP and GGPP is, then, pivotal in determining the synthesis of tocopherols and tocotrienols in plants. Additionally, α-tocotrienol can be converted to α-TC by the NADPH-dependent saturation of the prenyl chain, but the regulation of this reaction is still poorly understood [1,2]. 

Tocochromanol biosynthesis is divided between the chloroplast envelope and the plastoglobules (PG) connected to the thylakoid membranes. PGs are structures surrounded by a lipid monolayer that participate in the synthesis and storage of a variety of neutral lipids. PGs proliferate and increase in size in response to oxidative stress (e.g., during senescence, high-light exposure, or drought) when high levels of tocochromanols (especially tocopherols) are accumulated [9,15,16,17]. Tocopherols play a major role in maintaining the integrity of chloroplast membranes by quenching lipid peroxidation during senescence (associated with the conversion of chloroplasts into gerontoplasts) and as photoprotective agents against the excess of light. For the latter, α-TC associates to chloroplast membranes and contributes to photoprotection by quenching ^1^O_2_ generated by chlorophyll degradation products in photosystem II (PSII) through the formation of tocopherol quinone [2,12,18]. 

The biotechnological improvement of the VitE content in crop plants has become a major goal of plant metabolic engineering in the last few years [14,19]. The main attempts involved increasing the flux of the biosynthetic pathway (for example, by directly overexpressing genes involved in tocochromanol biosynthesis) or changing the tocochromanol profile to enrich the tissue in α-TC [20,21]. Many of the genes involved in the VitE metabolic pathway have been cloned and overexpressed by a stable transformation in multiple species, including *A. thaliana*, tobacco, canola, soybean, corn, lettuce, and tomato. The results showed different degrees of success depending on the gene used, the plant species, and the tissue where they were expressed [5,20]. Seeds have been the main target of the trials due to their naturally high content of γ-TC convertible to α-TC. Overexpression of the *VTE4* gene encoding γ-TMT allows the channelling of tocopherol biosynthesis towards α-TC, however, few attempts have been made to test whether this gene could be combined with other strategies to “push” the synthesis of precursors [3,11]. Poorly explored aspects of VitE biofortification strategies also include promoting tocochromanol sequestration (“pull” approaches), engineering of green tissues such as leaves (where changing the tocochromanol profile can negatively impact photosynthesis), or combined enrichment of several groups of related phytonutrients. In this work we performed agroinfiltration of *Nicotiana benthamiana* leaves to test the combination of “push” and “pull” strategies [22] aiming to increase VitE biosynthesis while promoting the formation of PG as appropriate storage structures where tocopherols can be accumulated in greater amounts. The results successfully demonstrate the feasibility of enriching edible green leaf crops in VitE (in particular, α-TC) and in other isoprenoid vitamins such as provitamin A carotenoids and phylloquinone (vitamin K1) [10,23].

## 2. Results and Discussion

### 2.1. VitE Content in Leaves Can Be Maximised by Enhancing the Biosynthetic Pathway in Combination with the Formation of Artificial Chromoplasts for Storage

We have previously shown that overexpression of the bacterial *crtB* gene in plant leaves grants a consistent accumulation of tocopherols (mainly γ-TC and α-TC) associated with the proliferation of PG, which is involved in the biosynthesis and storage of these isoprenoid compounds [10]). Leaf chromoplast formation and PG proliferation, in fact, also provide an improved environment for the activity of biosynthetic enzymes and a suitable space to sequester these metabolites [10,23]. We used *N. benthamiana* as a model platform because it is a fast-growing species particularly suitable for transient overexpression of multiple genes. First, a version of crtB equipped with a transit peptide (p-crtB) for efficient plastid targeting [23] was overexpressed in combination with single genes of the tocopherol biosynthetic pathway, namely *VTE1*, *VTE2*, *VTE3*, *VTE4*, *VTE5*, *VTE6*, and *tyrA* (a bacterial gene that catalyzes the formation of Tyr starting from prephenate and thus avoiding the feedback inhibition associated with this compound). The resulting pairs were named respectively 1B, 2B, 3B, 4B, 5B, 6B, and AB. In parallel, we also tested multiple gene combinations intended to maximize the production of α-TC, including *tyrA* + *VTE1* + *VTE2* + *VTE4* + *p-crtB* (A124B) and *VTE5* + *VTE1* + *VTE2* + *VTE4* + *p-crtB* (5124B). The accumulation of different tocopherol forms in leaves agroinfiltrated with these combinations was compared to that in leaves producing only p-crtB or a cytosolic version of the GFP protein as a control [23] (Figure 2).

The overexpression of *VTE2* or *VTE3*, combined with *p-crtB*, did not produce any relevant increase in total tocopherol content compared to that observed with only p-crtB (Figure 2A). This result diverges from that observed in other studies that reported that overexpression of *VTE2* or *VTE3* was sufficient to increase tocopherol levels in *N. benthamiana* leaves (i.e., in chloroplasts) and in multiple leafy plants such as *A. thaliana*, tobacco, and lettuce [14,24]. We speculate that these key steps might not be limited to increasing the levels of tocopherols in artificial leaf chromoplasts, while in chloroplasts they could have a more fundamental role. Notably, VTE2 and VTE3 protein localization in chromoplasts has not been confirmed yet, while they have been assigned to the inner envelope of chloroplasts. Changes in plastid ultrastructure and in enzyme localization could reduce either the access to precursors or the access to the downstream enzymes to the MPBQ and dMPBQ pools [25,26]. Furthermore, downstream genes such as *VTE1* and *VTE4* were found to be expressed at lower levels in leaves developing artificial chromoplasts compared to control leaf tissues harboring chloroplasts [10], so it is possible that a putatively higher supply of precursors due to *VTE2* or *VTE3* overexpression could not be efficiently converted into tocopherol end products.

Transient overexpression of *VTE1* in *N. benthamiana* leaves coexpressing p-crtB produced an increased level of total tocopherols, mainly supported by the increase in γ-TC (Figure 2A). In *A. thaliana,* the overexpression of *VTE1* in leaves under the control of a CaMV 35S promoter resulted in a sevenfold increase in the content of total tocopherols compared to wild-type plants, 81% of which was due to γ-TC with no significant increase in the α-TC level [27]. The excess of γ-TC, in fact, resulted in the saturation of VTE4, with some of the substrate not accessible to the enzyme [14,28]. Similarly, *VTE1* expressed together with *p-crtB* increases the amount of γ-TC threefold compared to p-crtB alone, enhancing total tocopherol levels without impacting the levels of α-TC. VTE1 efficacy might rely on the presence of increased numbers of PGs in artificial *N. benthamiana* chromoplasts [10]. In fact, the association of VTE1 to PG has been demonstrated through a variety of different techniques, including VTE1-fluorescence tagging, immunoelectron microscopy, immunoelectron tomography, and Western blotting of isolated chloroplast membrane fractions [29,30]. Endogenous VTE1 protein content was reported to increase during chromoplastogenesis in multiple species such as tomato, pepper, or the same p-crtB-expressing *N. benthamiana* leaves [10,15]. Furthermore, dMPBQ, the substrate of VTE1, accumulated in the PG of the *Arabidopsis vte1* mutant, providing evidence for the enzymatic activity of VTE1 at the PG, whereas tocopherols were found to accumulate mainly in the proliferating PG of artificial chromoplasts [10,28]. 

As in previous studies [31], the overexpression of *VTE4* together with p-crtB in *N. benthamiana* leaves did not change the total amount of tocopherols but resulted in the sole accumulation of α-TC (Figure 2). VTE4 protein was reported to be localized at the inner envelope of chloroplasts but to have high accessibility to the γ-TC pool present in the PG [30]. Chromoplast PG could be more efficient in the recruitment of enzymes involved in tocopherol biosynthesis, facilitating the role of VTE4 in a similar fashion to what was observed for carotenoid biosynthesis [26]. The combination of *VTE4* with *p-crtB* allows the production of leaf material with VitE amounts that are comparable to those found in soybeans [28].

The coexpression of *p-crtB* with genes required for tocopherol precursor supply, such as *VTE5*, *VTE6*, and *tyrA*, resulted in similar increases in the level of total tocopherols (1.3-fold) compared to *p-crtB* alone (Figure 2), suggesting that the availability of substrates (either phytol or Tyr) limits tocopherol synthesis in artificial chromoplasts. Consistently, the availability of phytol has already been determined as the most limiting factor in VitE biosynthesis through studies in cell cultures [32]. Furthermore, multigene combinations A124B and 5124B (i.e., when only one of the precursor’s branches is overexpressed) resulted in similarly increased levels of total tocopherols (likely due to the higher availability of the precursors) with the sole presence of α-TC due to the presence of the VTE4 gene able to convert the γ-TC pool (Figure 2). The success of these two combinations confirms that the entire pathway must be upregulated to obtain high α-TC levels. A combination of *VTE2*, *VTE1*, and *VTE4* had already been effectively tested to improve the α-TC levels in *Arabidopsis* and *N. tabacum* [14,20], however, our study shows that it is possible to further increase VitE accumulation by adding new steps to the combination that contribute to the availability of upstream precursors (i.e., phytol and Tyr). It is worth noting that, even if the tocopherol biosynthetic pathway could potentially compete for GGPP with the carotenoid and chlorophyll biosynthetic pathways, the enhanced accumulation of tocopherols did not substantially impact the levels of these photosynthetic pigments in chromoplasts (i.e., their levels were higher than those in the chloroplasts and were roughly like those of plants expressing only p-crtB) (Figure 2). Interestingly, p-crtB-mediated chromoplastogenesis in leaves results in an increased expression of genes involved in the synthesis of GGPP precursors [23], which might contribute to the support of tocopherol biosynthesis without competition for carotenoid and chlorophyll biosynthesis. Alternatively, a high level of tocopherols could protect against natural oxidation and degradation of carotenoids [33]. Indeed, the leaves showing enhanced levels of tocopherols (particularly α-TC) also displayed a higher antioxidant activity (Figure 2).

### 2.2. Physical Treatments Associated with PG Proliferation Can Be Combined with Metabolic Engineering for Additional Tocopherol Enhancement

A relationship between PG abundance and isoprenoid storage has been characterized in multiple plant species [9,10,15]. In particular, PG is the main storage site for isoprenoids (including tocopherols and carotenoids) in crtB-triggered artificial leaf chromoplasts [10]. We hypothesized, thus, that the use of physical treatments associated with increased PG proliferation could further enhance the accumulation of these metabolites. First, we tested the effect of a high-light treatment. Exposure to intense light stimulates the proliferation of PG by inducing high levels of thylakoidal membrane remodeling and lipids mobilization that correlate to the accumulation of PG core proteins such as FBN1a and FBN2 [34]. Also, the oxidative stress induced by high light, and the related production of ROS, have been shown to play a key role in facilitating chromoplast development in leaves providing a tool for a faster increase of sink (i.e., “pull”) capacity [10]. *N. benthamiana* plants growing under standard conditions (50 μmol photons·m^−2^·s^−1^ of white light, herein referred to as W50) were exposed for three days to tenfold higher light intensity (W500). Then, they were agroinfiltrated with the combinations 1B, 4B, 5B, and 5124B and left at W50 for four days (Figure 3). As expected, the W500 light treatment increased the levels of tocopherols and carotenoids compared to W50-grown samples (Figure 2). Interestingly, in 1B samples, the exposure to high light resulted in increased levels of α-TC compared to what was observed in plants grown under lower light conditions and expressing the same genes. It is possible that the high light treatment not only upregulates PG proliferation, but also the expression of tocopherol biosynthetic genes (e.g., *VTE4*) or enzymatic activity due to an increased demand for ROS scavenging in the plant that requires increased levels of highly antioxidant compounds such as α-TC. The enhancement would make the enzyme less prone to saturation and able to process the whole γ-TC pool, avoiding the limitations found when overexpressing only *VTE1.* This hypothesis is in line with what was suggested by Piller et al. [35], who reported that *VTE1* overexpression in *A. thaliana* under high light exposure yielded a threefold increase in α-TC. However, biological differences between plant species must be considered as they might affect the response to high irradiances.

To further demonstrate the role of PG in the improvement of tocopherol content in leaves, we then tested the effect of dark-induced senescence on the combinations 1B, 4B, 5B, 6B, and 5124B (Figure 4). We included VTE6 among the used combinations since, together with VTE5, it has a key role in the recycling of the phytol group from chlorophyll during senescence. In several plant species (e.g., barley and wheat), dark-induced senescence triggers rapid chloroplast degradation and α-TC and γ-TC increase, followed by a decline in the later stages [2]. It has been hypothesized that tocopherols might help maintain the integrity of chloroplast membranes during senescence and participate in the transformation of chloroplasts to gerontoplasts by controlling lipid peroxidation and the formation of oxidation products such as jasmonic acid, a signal molecule, able to modulate the expression of senescence-related genes (e.g., SAG12) [2,36]. Consistently, the increase of tocopherol levels in dark-incubated, senescent leaves, was very drastic (Figure 4). Particularly in 5124B, we detected a 25-fold increase in α-TC compared to nonsenescent *p-crtB* controls (Figure 4B). The effect could be explained by the direct correlation between chlorophyll degradation and tocopherol synthesis observed during senescence [37]. Senescent leaves naturally contain high tocopherol levels and show a naturally high expression of VTE5 and VTE6 which have a central role in chlorophyll mobilization during thylakoid dismantling. [38]. Moreover, during senescence, PGs decrease in number to merge into a few big structures where the products of chlorophyll degradation and thylakoid complex dismantling are deposited. Chromoplast senescence was reported and studied in several fruits and vegetables such as red bell pepper and citrus and it was associated with increased recruitment of proteins involved in membrane degradation, stress response, and scavenging compound biosynthesis [39]. Senescent leaves were reported to be the site of the highest expression of genes involved in the biosynthesis of secondary metabolites such as artemisinin [40]. Likely, *VTEs* overexpression would continue during the entire senescence process, contributing to high levels of tocopherols, readily accumulated in the super-sized PG present in gerontoplasts. 

While W500 exposure contributed to a higher level of carotenoids in leaves (Figure 3), dark incubation led to a slight reduction in the content of total carotenoids compared with non-senescent leaves overexpressing *p-crtB* (Figure 4). This effect is well characterized [41]. However, considering the aim of our study, it could be considered a hindrance as it does not allow the biofortification of multiple compounds. It works very well for tocopherols but not for carotenoids. Moreover, considering this study as a first step for the development of a commercial product, a senescent leaf would be hardly appealing to the consumer.

### 2.3. The Levels of Plastidial Isoprenoids Other than Tocopherols and Carotenoids Are Also Impacted by Tocopherol Pathway Engineering and Chromoplast Differentiation

In addition to carotenoids (including provitamin A β-carotene) and tocopherols (VitE), artificial leaf chromoplasts accumulate other plastidial isoprenoids [10]. Among them, plastochromanol-8 (PC-8), plastoquinone-9 (PQ-9), phylloquinone (vitamin K1), and α-tocoquinone (α-TQ) are particularly interesting due to their antioxidant properties and their metabolic connections with the tocopherol pathway (Figure 1). VTE3, for example, is responsible for converting 2,3-methyl-6-solanesyl-1,4-benzoquinol (MSBQ) to plastoquinol, which is then oxidized to PQ-9 or converted into PC-8 by VTE1 [42]. α-TQ is a product that results from the antioxidant activity of α-TC. Lastly, phylloquinone is synthesized from a chorismate-derived moiety and phytyl-PP [43]. We investigated the levels of these plastidial isoprenoids in the same leaves expressing the 1B, 2B, 3B, 4B, 5B, 6B, AB, A124B, and 5124B combinations used for tocopherol and carotenoid measurements (Figure 3).

The levels of α-TQ fluctuated among the different samples, however, none of the tested combinations showed any substantial impact compared to the p-crtB controls (Figure 5). It is likely that exposing the plant to a high light stress or any other treatment resulting in an increased oxidation of α-TC might lead to stronger changes in α-TQ levels. The formation of α-TQ, in fact, is relevant as a response to photooxidative and low-temperature stress where oxidation of α-TC forms α-tocopheroxyl radical which decomposes to a more stable α-TQ but is also dependent on α-TC availability [44] *Arabidopsis* vte4 mutants which accumulate γ-TC instead of α-TC in their leaves do not show any trace of α-TQ [45]. In our experiments, however, we did not detect a significant increase in α-TQ in leaves overexpressing *VTE4* and *p-crtB*, meaning that, in control conditions, the direct conversion of α-TC (highly present in *VTE4*-overexpressing leaves) to α-TQ, is limited. 

The overexpression of *VTE1* resulted in an increased amount of PC-8, as expected (Figure 1). Conversion of PQ-9 to plastoquinol can happen in response to photooxidation and thylakoid degradation in plant leaves or tomato and pepper fruits [26,42,45]. Likely, a high basal level of available plastoquinol in crtB chromoplasts might provide the substrate for VTE1 function while the increased PG number provides the synthesis and storage space [9,10]. Surprisingly, the overexpression of *VTE3* recovered PQ-9 amounts bringing them to a similar level to those of a normal green leaf (Figure 5). Normally PQ-9 levels in chromoplasts are low because they are associated with thylakoidal membranes that undergo a degradation process during chromoplastogenesis [42]. Supporting our previous theory, the removal of plastoquinol in this combination resulted also in a lower amount of PC-8 compared to leaves overexpressing only p-crtB. It is likely that exposure to high light conditions could further increase the content of this metabolite in chromoplasts if performed before the start of the chloroplast conversion. Additionally, leaves overexpressing *VTE2* are the ones showing the lowest levels of PQ-9 since the encoded HPT enzyme readily converts HGA to MPBQ, removing the substrate for MSBQ synthesis (Figure 1). Finally, the overexpression of *VTE5* and *VTE6* increased phylloquinone levels while *tyrA* had no relevant effect on any compound analyzed. Phytol, phytyl-P and phytyl-PP, but not Tyr, are fundamental for phylloquinone biosynthesis (Figure 1). Accordingly, *Arabidopsis vte5* and *vte6* mutants show a drastic reduction of phylloquinone content and high sensibility to photooxidative stress [25,46]. Samples overexpressing the 5124B combination showed significantly lower levels of phylloquinone, compared to the p-crtB control as the phytol precursors were completely channeled to α-TC. Peroxisome proliferation, moreover, is frequent during chromoplast formation in fruit in response to changes in the activity of the antioxidant system [47]. Thus, it cannot be excluded that the observed enrichment in vitamin K1 levels results from the combined effect of higher availability of substrate and of peroxisomes (where part of the phylloquinone biosynthetic pathway occurs).

## 3. Conclusions

The overexpression of genes involved in VitE biosynthesis has been used as a strategy to increase the levels of this important vitamin in plant tissues in multiple studies. In this work we have successfully characterized the effect of mixing this “push” approach with a “pull” strategy based on the promotion of chromoplast formation and particularly PG proliferation in leaves, demonstrating its efficacy to improve the nutritional content of green tissues. This is a groundbreaking result since it achieved three main objectives of VitE biofortification: (1) increased tocopherol synthesis, (2) maximized the proportion of α-TC, and (3) improved tocopherol sequestration and storage. Aside from the novelty of successfully implementing a combination of “push” and “pull” approaches, our study shed a light on what the steps of VitE biosynthesis are to be considered when planning a biofortification strategy, not only for their role in the synthesis of tocopherols but also for their effect in the accumulation of other related isoprenoid compounds of nutritional interest. This was possible thanks to the use of the agroinfiltration technique, which allows a fast overexpression of multiple gene combinations in allocated plant tissues (i.e., leaves) and hence does not compromise overall plant fitness even when the changes in the agroinfiltrated area impair essential functions such as photosynthesis. Different systems for transient gene expression [48] should allow the use of the gene combinations reported here to obtain green leafy vegetables simultaneously enriched in VitE, provitamin A carotenoids, and vitamin K1 once photosynthesis is dispensable (i.e., right before harvest). This biotechnological strategy can be combined with exogenous treatments such as high light exposure that could be applied in the context of vertical farming to accelerate vitamin production and accumulation [10] and hence reduce the harvest time, eventually increasing productivity. Future characterization of the subcellular localization sites, protein interaction partners, and mechanisms of action of all the enzymes involved in these metabolic pathways would be the key to the more precise design of metabolic engineering approaches and treatments that promote biofortification with a minimum impact on plant fitness. Finally, it is worth remembering that biofortification (i.e., the production of high levels of nutritionally relevant metabolites in plant food products) improves bioavailability and has a lower environmental impact compared to the use of dietary supplements or food fortification with the same compounds obtained by chemical synthesis.

## 4. Materials and Methods

### 4.1. Plant Material and Growth Conditions

*Nicotiana benthamiana* plants were grown in a greenhouse under standard long-day conditions (LD, 14 h light at 26 ± 1 °C and 10 h dark at 21 ± 1 °C) as described previously [23,49]. For the intense light experiments, plants were grown under standard conditions and then moved to an ARALAB 600 growth chamber where they were maintained for three days in two different light conditions: 50 μmol photons·m^−2^·s^−1^ (named W50) and 500 μmol photons·m^−2^·s^−1^ (W500) under a long day (16 h light and 8 h dark) photoperiod. After agroinfiltration, plants were all maintained in the W50 condition. For the dark induced senescence experiment, 4 days after the agroinfiltration, leaves where detached and kept for 10 days on moist Whatman paper in petri dishes fully covered with aluminum foil. Plant tissue of interest for further analysis was detached, frozen in liquid nitrogen, freeze-dried for 24 h in a laboratory freeze drier (A 2–4 LD plus, CHRIST, Osterode am Harz, Germany), and stored at −80 °C until further analysis.

### 4.2. Gene Constructs

The crtB version used for this study (35S:p-crtB-GFP-pGWB405) was generated as described [23]. Transcript encoding for the VTE1 protein was obtained from *A. thaliana* seeds [10,29] and VTE2, VTE3, VTE4, VTE5, and VTE6 proteins from *A. thaliana* leaves. For the sequence encoding for the tyrA protein, a culture of *E. coli* was used as a template. Primers used for this amplification are listed in Table A1. PCR products were cloned using the Gateway system first into plasmid pDONR-207 and then into plasmid pGWB454 provided with an RFP fluorescent tag [50] to generate 35S:VTE1-pGWB454, 35S:VTE2-pGWB454, 35S:VTE3-pGWB454, 35S:VTE4-pGWB454, 35S:VTE5- pGWB454, 35S:VTE6-pGWB454, and 35S:tyrA-pGWB454. For agroinfiltration assays, the second or the third youngest leaves of 4–5-week-old *N. benthamiana* plants were infiltrated in the abaxial part of the leaves. *Agrobacterium tumefaciens* GV3101 strains were transformed with the appropriate constructs and grown on Luria-Bertani (LB) agar plates with the corresponding antibiotics at 28 °C for 3 days. A single PCR-confirmed colony per construct was inoculated in 15 mL antibiotic-complemented LB media and incubated for 3 days at 28 °C in continuous agitation before performing agroinfiltration. The assay was carried out using cells from fresh cultures at an optical density of 600 nm of 0.5 in infiltration buffer (10 mM MES, 10 mM MgCl_2_, and 150 μM acetosyringone, pH = 5.5). Cultures were mixed in identical proportions when agroinfiltrating several constructs. Gene silencing was prevented by coagroinfiltration with an agrobacterium strain EHA101 carrying the helper component protease (HcPro) of the watermelon mosaic virus (WMV) in plasmid HcProWMV-pGWB702 (a kind gift of Juan José López-Moya and Maria Luisa Domingo-Calap (CRAG-Barcelona, Spain)).

### 4.3. Metabolite Analyses

Leaf carotenoids, chlorophylls, and tocopherols were extracted in 2 mL Eppendorf tubes from 4 mg of freeze-dried leaf tissue, using 375 µL of methanol as the extraction solvent and a 25 µL of 10% (*w*/*v*) solution of canthaxanthin in chloroform (Sigma) as the internal standard. Tissue was lysed by adding 4 mm glass beads and grinding for 1 min at 30 Hz in a TissueLyser II (QIAGEN, Venlo, Netherlands) and the extraction was carried out by adding 400 µL of Tris-NaCl pH 7.5 and 800 µL of chloroform. Thoroughly mixed samples were centrifuged for 5 min at 13,000 rpm at 4 °C and the organic phase was transferred into a new tube and evaporated using a SpeedVac system (Eppendorf Concentrator plus, Hamburg, Germany). The extracted metabolites were then completely redissolved in 200 µL of acetone, filtered with 0.2 µm filters into amber-colored 2 mL glass vials and a 10-µL aliquot of each sample was then injected onto an Agilent Technologies 1200 series HPLC system (Agilent Technologies, Santa Clara, CA, USA). A C30 reverse-phase column (YMC Carotenoid, 250 × 4.6 mm × 3 µm, YMC CO., Kyoto, Japan) was used, with three mobile phases consisting of methanol (solvent A), water/methanol (20/80 *v*/*v*) containing 0.2% ammonium acetate (*w*/*v*) (solvent B), and tert-methyl butyl ether (solvent C). Metabolites were separated following the following gradient: 95% A, 5% B isocratically for 12 min, a step-up to 80% A, 5% B, 15% C at 12 min, followed by a linear gradient up to 30% A, 5% B, and 65% C by 30 min. The flow rate was maintained at 1 mL/min. The HPLC equipment was coupled to a Photometric Diode Array (PDA) detector (Santa Clara, CA, USA) allowing the detection of the full UV-visible absorption spectra of the different metabolites. Peak areas of chlorophylls at 650 nm and carotenoids at 472 nm were determined using Agilent ChemStation software. A fluorescence detector at 330 nm was used for tocopherol identification. The quantification of the compounds of interest was done by using a concentration curve built with a commercial standard (Sigma-Aldrich, Steinheim, Germany) [51]. For the rest of isoprenoid metabolites, approximately 7 mg of freeze-dried tissue were mixed with 500 µL of THF:MeOH (Analytical grade, Normapur) 1:1 buffered with 10% of water (*v*/*v*), thoroughly mixed, centrifuged, transferred to an amber vial, and injected into a Waters Acquity UPLC™ (Milford, MA, USA) coupled to a Waters Synapt G2 MS QTOF equipped with an atmospheric pressure chemical ionization (APCI) source. Prenyllipids were separated on an Acquity BEH C18 column (50 × 2.1 mm, 1.7 µm) under the following conditions: Solvent A = water; Solvent B = MeOH; 80–100% B in 4.0 min, 100% B for 2.5 min, re-equilibration at 80% B for 2.0 min. The flow rate was 500 µL/min, and the injection volume was 2.5 µL. Standards of HPLC grade (≥99.5%) were purchased from Sigma-Aldrich. PQ-9 and PC-8 standards were purified in house [52]. 

### 4.4. Antioxidant Capacity

Metabolite extracts prepared as described above were diluted in 400 µL of diethyl-ether and saponified by adding 100 µL of 10%(*w*/*v*) KOH in methanol to avoid interference from chlorophylls. Samples were left shaking for 30 min at 4 °C and then diluted with 400 µL of milliQ water before centrifugation for 5 min at 13,000 rpm and 4 °C. The upper phase was collected, dried in a SpeedVac (Eppendorf Concentrator plus, Hamburg, Germany), and resuspended in 200 µL of acetone. ABTS assay to calculate the total antioxidant capacity of the mixture was carried out as described. Briefly, ABTS•+ was prepared by adding to a 7 mM solution of ABTS (Sigma-Aldrich) and APS to a final concentration of 2.45 mM. The solution was then diluted with water to reach an absorbance of 0.700 ± 0.020 at 30 °C. Next, 100 µL of the diluted sample were added to 1 mL of ABTS•+, and absorbance at 734 nm was measured after 4 min. The obtained values were plotted against a standard curve calculated with increasing 6-hydroxy-2,5,7,8-tetramethylchroman-2-carboxylic acid (Trolox) concentrations (11 µM, 22 µM, 44 µM, 88 µM) to calculate the antioxidant activity as µM of Trolox equivalents [49,53].

## Figures and Tables

**Figure 1 metabolites-13-00193-f001:**
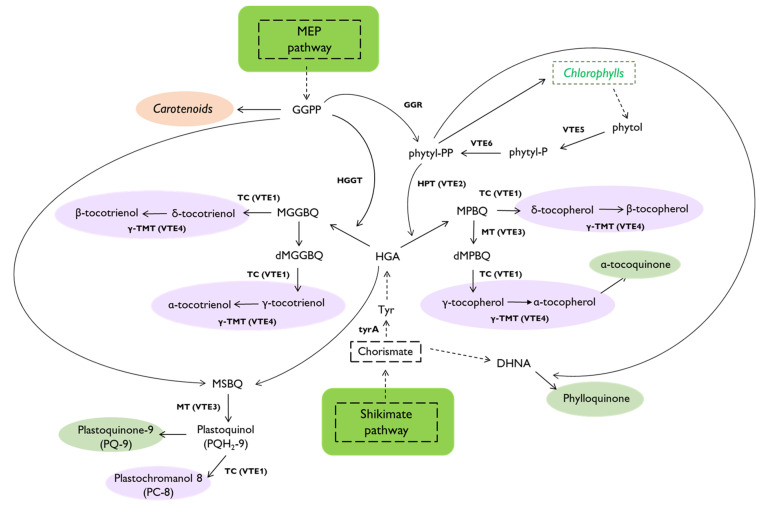
Schematic representation of tocochromanol biosynthetic pathways. Carotenoids are encircled in orange, chromanols in purple, and quinones in green. Dotted lines indicate multiple-step pathways where all the reactions are not represented. See text for acronyms.

**Figure 2 metabolites-13-00193-f002:**
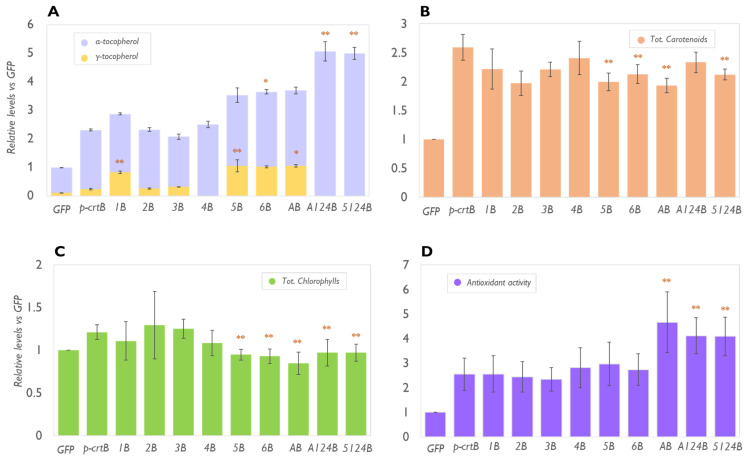
Genes promoting tocopherol biosynthesis and chromoplast differentiation can be combined to increase total VitE levels. Relative levels of tocopherols (**A**), total carotenoids (**B**), and total chlorophylls (**C**) in leaves overexpressing *p-crtB* alone or in combination with genes involved in tocochromanol biosynthesis (see text for acronyms). (**D**) Antioxidant activity of plant extracts obtained from leaves overexpressing the same gene combinations. Data are represented relative to the amounts measured in GFP samples (set to 1) and correspond to the mean and standard deviation of n = 3 different leaves. Orange asterisks mark statistically significant differences between each combination and p-crtB samples (*t*-test * *p* < 0.05, ** *p* < 0.01).

**Figure 3 metabolites-13-00193-f003:**
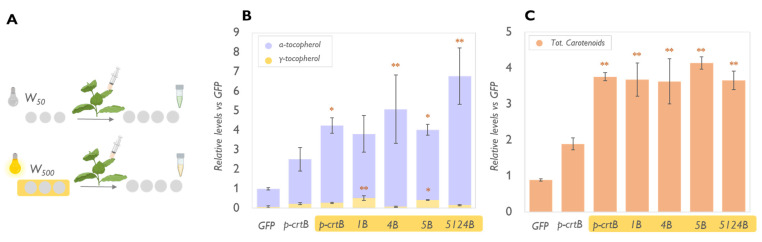
High light treatment enhances tocopherol and carotenoid accumulation in leaves overexpressing *p-crtB* and tocochromanol biosynthesis genes. (**A**) Schematic representation of the experimental design. The small Eppendorf icons symbolize the day of sample collection. Levels of tocopherols (**B**) and carotenoids (**C**) in leaves from the agrofiltrated plants are represented relative to the amounts measured in nontreated GFP samples (set to 1) and correspond to the mean and standard deviation of n = 3 different leaves. Yellow boxes highlight W500-treated samples. See text for acronyms corresponding to gene combinations. Orange asterisks mark statistically significant differences compared to values observed in the control (W50) p-crtB samples (*t*-test * *p* < 0.05, ** *p* < 0.01).

**Figure 4 metabolites-13-00193-f004:**
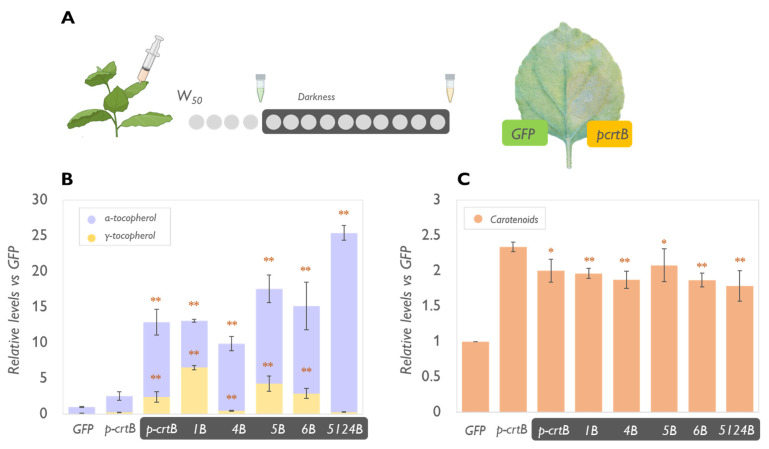
Dark-induced senescence enhances tocopherol accumulation but slightly decreases carotenoid content. (**A**) Schematic representation of the experimental design and representative picture of an agroinfiltrated leaf immediately before sampling. Leaves were agroinfiltrated with the indicated combinations of constructs (see text for acronyms) and after 4 days they were detached and incubated in the dark for 10 additional days. The small Eppendorf icons symbolize the days of sample collection. Levels of tocopherols (**B**) and carotenoids (**C**) are represented relative to the amounts measured in the control (nonsenescent) GFP samples (set to 1) and correspond to the mean and standard deviation of n = 3 different leaves. Dark-incubated samples are highlighted in dark boxes. Orange asterisks mark statistically significant differences compared to values observed in the control (W50-grown leaves sampled 4 days after the infiltration) p-crtB samples (*t*-test * *p* < 0.05, ** *p* < 0.01).

**Figure 5 metabolites-13-00193-f005:**
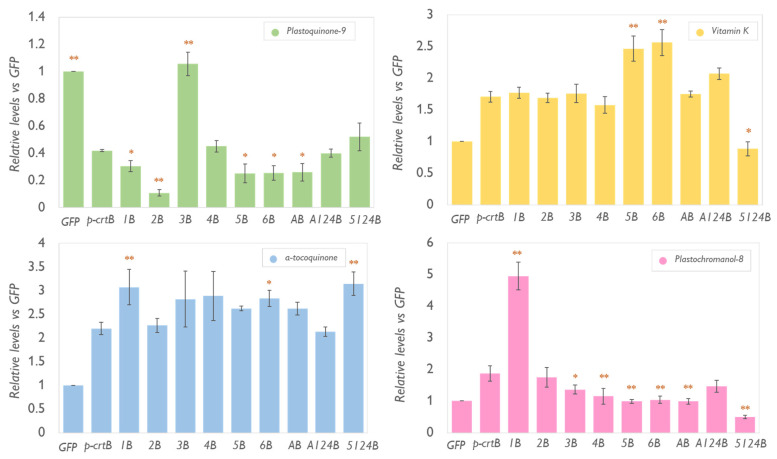
Overexpression of genes involved in tocochromanol biosynthesis increase levels of other plastidial isoprenoids derived from shared metabolic pathways in chromoplasts. Levels of plastidial isoprenoids in leaves overexpressing the different enlisted combinations (see text for acronyms) are represented relative to the amounts measured in GFP samples (set to 1) and correspond to the mean and standard deviation of n = 3 different leaves. Orange asterisks mark statistically significant differences between each combination and p-crtB samples (*t*-test * *p* < 0.05, ** *p* < 0.01).

## Data Availability

All the data supporting the findings of this study are included in this article and the Appendix A.

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
