# Peer review of "Nutritional Enrichment of Plant Leaves by Combining Genes Promoting Tocopherol Biosynthesis and Storage"

_metabolites, 2023, doi:10.3390/metabo13020193_

Round 1

Reviewer 1 Report

This ms reports on a study to investigate the coexpression of crtB with genes involved in tocopherol biosynthesis and its potential in vitE accumulation in N.benthamiana leaves.

The work aimed to development and improvement of vitE content using Push and Pull strategies. The capacity of crtB gene to induce chromoplastogenesis in leaves was combined with the transient overexpression of multiple genes of the tocochromonal biosynthetic pathways

Specific comments:

The biosynthetic pathway (lines 65-94) should be succinctly explained. 

The materials and methods section needs to be elaborated. Not as previously described. As previously described etc.

The paper claims that stable transformations are time-consuming for biofortification experiments. But the results presented in this study need to be confirmed in stably transformed plants. n=3 for transient expression is low to have statistically significant results.

Author Response

Referee: 1

This ms reports on a study to investigate the coexpression of crtB with genes involved in tocopherol biosynthesis and its potential in vitE accumulation in N.benthamiana leaves.

The work aimed to development and improvement of vitE content using Push and Pull strategies. The capacity of crtB gene to induce chromoplastogenesis in leaves was combined with the transient overexpression of multiple genes of the tocochromonal biosynthetic pathways

Point 1: The biosynthetic pathway (lines 65-94) should be succinctly explained.

Response 1: The biosynthetic pathway description has been shortened while maintaining the fundamental information about the chemical reactions involved.

Point 2: The materials and methods section needs to be elaborated. Not as previously described. As previously described etc.

Response 2: The materials and methods section has been accordingly expanded to provide a detailed explanation of all the techniques used in this work.

Point 3: The paper claims that stable transformations are time-consuming for biofortification experiments. But the results presented in this study need to be confirmed in stably transformed plants. n=3 for transient expression is low to have statistically significant results.

Response 3: The scope of this paper was to test the possibility to combining “push” and “pull” strategies using multiple gene combinations. While generating stably transformed plants with the best combinations could be a logical continuation of the work, the differentiation of artificial chromoplasts by (crtB overexpression) prevents photosynthesis and is meant to be applied once the photosynthetic activity of the plant is dispensable (i.e., just before harvest). Stable transformants overexpressing crtB at the required levels are just not viable.

Reviewer 2 Report

In this paper, the authors combined co-expression of genes involved in tocopherol biosynthesis and found improved nutritional content in Nicotiana benthamiana leaves. The work is a significant contribution to the field and is well-organized and well-written. However, a couple of minor comments should be addressed.

The introduction needs to be improved by clearly showing what knowledge gaps are identified from literature review. And how the authors connect the knowledge gaps to their research goals. Please reason both the novelty and the relevance of your research goals, as has not been sufficiently highlighted in the current version. 

Figures should be appropriately described and labeled. Please explain the meaning of different colors in Figure 1. Please add the unit in y-axis for figures 2, 3, 4, and 5.

The discussion part for this manuscript can be improved. The discussion section is the main part of a paper. It should not just summarize the key results of the study, but to highlight the insights and the applicability of your results for future work. Several important questions shall be addressed: What research gaps can the study help to solve? Who benefits from the improvements? What’s next? A focus discussion would also enable the authors to state the contribution of the paper more clearly. 

Author Response

Referee: 2

In this paper, the authors combined co-expression of genes involved in tocopherol biosynthesis and found improved nutritional content in Nicotiana benthamiana leaves. The work is a significant contribution to the field and is well-organized and well-written. However, a couple of minor comments should be addressed.

Point 1: The introduction needs to be improved by clearly showing what knowledge gaps are identified from literature review. And how the authors connect the knowledge gaps to their research goals. Please reason both the novelty and the relevance of your research goals, as has not been sufficiently highlighted in the current version.

Response 1: We thank the reviewer for the comment and the suggestion. We have rewritten the last paragraph of the Introduction section to address the reviewer questions. Additionally, the Conclusion sections has been expanded to cover these issues.

Point 2: Figures should be appropriately described and labeled. Please explain the meaning of different colors in Figure 1. Please add the unit in y-axis for figures 2, 3, 4, and 5.

Response 2: The figure 1 has been modified to have a less confusing color scheme. In the revised version, the three main classes of metabolites presented in this work have been highlighted with different colors as explained in the figure legend. The data presented in the rest of figures show relative levels and, as such, do not have a unit. To better clarify this point, we have indicated in the figure legends that data are represented relative to the amounts measured in GFP samples (which were set to 1).

Point 3: The discussion part for this manuscript can be improved. The discussion section is the main part of a paper. It should not just summarize the key results of the study, but to highlight the insights and the applicability of your results for future work. Several important questions shall be addressed: What research gaps can the study help to solve? Who benefits from the improvements? What’s next? A focus discussion would also enable the authors to state the contribution of the paper more clearly.

Response 3: As indicated in our Response 1, Conclusion sections has been expanded to cover these issues.

Reviewer 3 Report

The manuscript is noteworthy,  the experimental design and develop is consistent and structured according to the objective.

However, I suggest a revision of some points:

(1) Abstract: Some biosynthetic genes (line 27) need to be clarified.

(2) Introduction: please add the relevant functions of bacterial gene crtB as well as its research progress and research prospects.

(3) It is desirable to provide some pictures of cytological observations demonstrating the proliferation of PG (line 27).

(4) Since the paper only measured the relative content of vitamin E, etc., could the authors provide other evidence of the effectiveness of the test, such as the expression level of the genes?

Author Response

Referee: 3

The manuscript is noteworthy, the experimental design and develop is consistent and structured according to the objective. However, I suggest a revision of some points:

Point 1: Abstract: Some biosynthetic genes (line 27) need to be clarified.

Response 1: As suggested by the reviewer, line 27 has been clarified and genes relevant for this work have been enlisted.

Point 2: Introduction: please add the relevant functions of bacterial gene crtB as well as its research progress and research prospects.

Response 2: Because the Introduction is quite long, we prefer to introduce the crtB gene in the Results and Discussion section, providing sufficient context to understand why we use it in the frame of the current work. Further information on crtB functions and progress can be found in the indicated references [10, 22, 23].

Point 3: It is desirable to provide some pictures of cytological observations demonstrating the proliferation of PG (line 27).

Response 3: We do not have any TEM observation on leaves exposed to high irradiances. However, we base our statements on bibliographic data. We have introduced a brief line better explaining the link between high light exposure and PG development (Lines 273-277). Moreover, to use a more neutral term we changed “inducing” for “associated with” in line 264.

Point 4: Since the paper only measured the relative content of vitamin E, etc., could the authors provide other evidence of the effectiveness of the test, such as the expression level of the genes?

Response 4: We appreciate the concern. The work was planned to be a study on VitE accumulation when overexpressing genes involved in VitE biosynthesis. We were able to confirm through the compounds accumulated, that our overexpression was working (e.g., the presence of sole α-tocopherol when overexpressing VTE4). Besides, previous work of our group on the optimization of the agroinfiltration protocol demonstrated that all the transgenes cloned in the vectors that we use are overexpressed (albeit with different profiles) for 1 to 5 days following the agroinfiltration of the constructs.

Round 2

Reviewer 1 Report

The authors responded to the points raised. The manuscript can be accepted in its present form.